# A Multidisciplinary Approach to Complex Dermal Sarcomas Ensures an Optimal Clinical Outcome

**DOI:** 10.3390/cancers14071693

**Published:** 2022-03-26

**Authors:** Hannah Trøstrup, Amir K. Bigdeli, Christina Krogerus, Ulrich Kneser, Grethe Schmidt, Volker J. Schmidt

**Affiliations:** 1Department of Plastic Surgery and Breast Surgery, Zealand University Hospital (SUH) Roskilde, University of Copenhagen, 4000 Roskilde, Denmark; christina.krogerus@regionh.dk (C.K.); vosc@regionsjaelland.dk (V.J.S.); 2Department of Plastic Surgery and Burns Treatment, Rigshospitalet, University of Copenhagen, 2100 Copenhagen, Denmark; grethe.schmidt@regionh.dk; 3Department of Hand, Plastic, and Reconstructive Surgery, BG Trauma Center Ludwigshafen, Heidelberg University, D-67071 Ludwigshafen, Germany; amir.bigdeli@bgu-ludwigshafen.de (A.K.B.); ulrich.kneser@bgu-ludwigshafen.de (U.K.)

**Keywords:** primary dermal sarcoma, reconstructive surgery, plastic surgery, microsurgery

## Abstract

**Simple Summary:**

Knowledge of cutaneous or dermal sarcomas is crucial due to the initially peaceful appearance of skin tumours, causing delays in accurate diagnoses. They can easily be mistaken for benign skin appendages or moles. This clinical entity spans from low-grade tumours with practically no metastatic potential to highly aggressive neoplasms with substantial morbidity. Initial adequate resection is the absolute mainstay of successful therapy. Extensive tissue defects after resection necessitate a multidisciplinary approach involving reconstructive surgical planning from onset. Clinical and morphological types of dermal sarcomas are presented together with examples of cases undergoing interdisciplinary treatment and advanced microsurgical reconstructive therapy. Clinicians should be aware of the types of dermal sarcomas; their specific clinical presentations and growth patterns; as well as the necessity of clear surgical margins and the significant progress in reconstructive surgery, which gives new hope for future management of complex dermal sarcomas.

**Abstract:**

Primary dermal sarcomas (PDS) belong to a highly clinically, genetically and pathologically heterogeneous group of rare malignant mesenchymal tumours primarily involving the dermis or the subcutaneous tissue. The tumours are classified according to the mesenchymal tissue from which they originate: dermal connective tissue, smooth muscle or vessels. Clinically, PDS may mimic benign soft tissue lesions such as dermatofibromas, hypertrophic scarring, etc. This may cause substantial diagnostic delay. As a group, PDS most commonly comprises the following clinicopathological forms of dermal sarcomas: dermatofibrosarcoma protuberans (DFSP), atypical fibroxanthoma (AFX), dermal undifferentiated pleomorphic sarcoma (DUPS), leiomyosarcoma (LMS), and vascular sarcomas (Kaposi’s sarcoma, primary angiosarcoma, and radiation-induced angiosarcoma). This clinical entity has a broad spectrum regarding malignant potential; however, local aggressive behaviour in some forms causes surgical challenges. Preoperative, individualised surgical planning with complete free margins is pivotal along with a multidisciplinary approach and collaboration across highly specialised surgical and medical specialties. The present review gives a structured overview of the most common forms of dermal sarcomas including surgical recommendations and examples for advanced reconstructions as well as the current adjunctive medical treatment strategies. Optimal aesthetic and functional outcomes with low recurrence rates can be achieved by using a multidisciplinary approach to complex dermal sarcomas. In cases of extended local tumour invasion in dermal sarcomas, advanced reconstructive techniques can be applied, and the interdisciplinary microsurgeon should be an integral part of the sarcoma board.

## 1. Introduction

Primary dermal sarcomas are rare. Subtypes are classified after the mesenchymal tissue in which they arise.

Dermatofibrosarcoma protuberans (DFSP), atypical fibroxanthoma (AFX) and dermal undifferentiated pleomorphic sarcoma (DUPS) evolve from connective tissues. Cutaneous leiomyosarcomas originate from the smooth muscle cells of the skin, and Kaposi’s sarcoma (KS) and angiosarcomas (AS) originate from vessels. In order to confirm the diagnosis, assess the extent and stage of the sarcoma and guide biopsies, radiological imaging (CT or MRI) is performed. Surgery is the first line of treatment in the majority of sarcoma cases. In Europe, the Fédération Nationale des Centres de Lutte Contre le Cancer (FNCLCC) grading system is generally used to describe soft tissue sarcomas. It distinguishes three grades of malignancy and includes the assessment of tumour differentiation, necrosis and mitotic count [1].

Complex primary dermal sarcomas requiring large tissue resections are treated in a multidisciplinary approach with participation from oncology, pathology, internal medicine, dermatology, orthopaedic surgery, plastic surgery, and general surgery. In cases of unresectable recurrence or metastatic spread, systemic medical treatment or adjunctive radiotherapy (RT) is offered. However, as dermal sarcomas as a group are very heterogenous, strict guidelines are challenging to obtain. An individual approach and thorough discussion in a highly specialised sarcoma board are advised.

Basic knowledge of the surgical management of these rare but potentially life-threatening diseases, which often present as benign lesions of the skin, is crucial for physicians working in primary health care as well as in more specialised clinical entities at hospitals.

## 2. Types of Dermal Sarcomas

### 2.1. Dermatofibrosarcoma Protuberans

Dermatofibrosarcoma protuberans (DFSP) is the most common deep dermal neoplasm, comprising 6% of all soft tissue sarcomas. It usually involves the dermis and the subcutaneous layer. The peak incidence is observed in the second to fourth decades of life [2]. It exhibits a predisposition to local recurrence and destruction, growing with finger-like projections. Importantly, there is potential for evolution to higher grade sarcoma [3]. Besides the subclinical, infiltrative and extensive behaviour, however, the overall risk of metastasis is low unless it has fibrosarcomatous differentiation. It presents as a slowly progressing, painless, pink or violet plaque, usually located on the body or the limbs [4] and mimics benign lesions such as scarring or dermatofibromas. This causes clinical pitfalls by potentially delaying accurate diagnosis.

Histologically, DFSP is characterised by storiform, monomorphic spindle cell proliferation. CD34-positive cells can be used to determine DFSP from cellular dermatofibromas [5].

A genomic reciprocal translocation in t (17;22) (q22;q13) characterises DFSP. It results in fusion of the promoter of the collagen type Iα1 gene (COL1A1) and the platelet-derived growth factor B-chain (PDGFB) genes, which ultimately contribute to continuous autocrine production of local PDGF, promoting tumour growth [6,7]. The fusion gene can be detected by real-time PCR or fluorescent in situ hybridisation. A pigmented DFSP is referred to as a Bednar tumour. This rare variant, which comprises 1–5% of all DFSPs, has an additional presence of melanocytic dendritic cells and displays intermediate malignancy [8].

The gold standard for DFSP treatment is surgical resection with negative margins of a minimum of 2–3 cm including the fascia [9]. A reconstructive surgeon should always be part of the surgical planning when clear margins appear difficult to obtain. In cases of metastatic disease, radiation may be applied, but clinical outcomes are often disappointing. Systemic targeted biological therapy with tyrosine kinase inhibitors, such as Imatinib mesylate shows promising potential as targeted antitumour therapy. The FDA granted approval for the treatment of DFSP with this multikinase inhibitor in the PDGFRB signalling pathway in 2006. Clinical studies revealed that Imatinib is active in DFSP and fibrosarcomatous DFSP. The objective response rate approached 50% in two distinct phase II trials of Imatinib (400 to 800 mg daily) [10]. Overall, the 5-year survival of DFSP is 99.2%. However, insufficient initial surgery with radical excision with safety margins can cause poor prognosis and aggressive local tumour or bone invasion. In these cases, and in cases with fibrosarcomatous differentiation, diagnostic CT/MRI/PET-CT scan may reveal metastatic spread to the lungs [11,12]. An example of a complex case of DFSP and the reconstruction of the surgical defect is shown below (Figure 1).

### 2.2. Atypical Fibroxanthomas

Atypical fibroxanthomas (AFX) are low-grade dermal sarcomas. They often present as rare, exophytic, dermal ulcerative, haemophilic, and rapidly growing lesions derived from myofibroblasts. They can be mistaken for pyogenic granulomas because of the similar clinical presentation: a single, pinkish or red, polypoid, often crusted neoplasm. Exposure to ultraviolet light predisposes to the development of AFX, which is accordingly observed as a rapidly growing nodule with an ultimate size of 2–3 cm with a predilection for the head and neck region of elderly patients. It is regarded as a dermal, spindle-cell fibrohistiocytic tumour that only rarely reoccurs or metastasises. There seems to be some evidence that AFX is a superficial, less aggressive variant or precursor of dermal undifferentiated pleomorphic sarcoma as they share some similarities histologically [14]. The tissue is characterised by highly atypical cells with nuclear enlargement, pleomorphism, hyperchromasia, and a high mitotic rate [15].

Like in dermal undifferentiated pleomorphic sarcomas, 9p and 13q deletions are observed [16]; however, an absence of K-ras and H-ras mutations in AFX may explain the differing malignant potential of the two types of sarcomas [17]. Besides genetic alterations, known risk factors to the development of AFX are exposure to sunlight, immunosuppression and previous radiation therapy. Adequate initial surgical resection margins are mandatory. Radiotherapy may be a relevant adjuvant therapy but should never replace surgery.

### 2.3. Dermal Undifferentiated Pleomorphic Sarcoma

Dermal undifferentiated pleomorphic sarcoma (DUPS), previously referred to as malignant fibrous histiocytoma, is an undifferentiated pleomorphic variant with overlapping features of AFX. Histopathology shows pleomorphic areas of anaplastic spindle cells, often mixed with bizarre giant cells, in addition to an ordinary fascicular pattern of elongated smooth muscle-like cells. It stains positive for CD10 and actin–α smooth muscle [18].

The lesions are distinguished from AFX by a potential to invade deeper tissues and lymphoid or neural invasion. They exhibit local aggressive behaviour and the ability for distant metastasis. Clinically, a rapidly growing tumour with a strong predilection for sun-damaged skin on the head is observed [19].

Cytogenetically, 50–70% of AFX and DUPS have NOTCH1/2 and FAT1 mutations [20], in addition to frequent TERT promoter mutations [21], the latter leading to increased telomerase expression and continuous cell proliferation [22]. Assessed by next-generation genome wide sequencing and immunohistochemistry, UV-induced *p53* mutations as well as CCND1/CDK4 changes were observed in AFX and DUPS and may contribute to the tumorigenesis [23]. In a retrospective study involving 319 patients with diagnosed termed atypical fibroxanthoma, malignant fibrous histiocytoma, pleomorphic dermal sarcoma, and subfascial undifferentiated pleomorphic sarcoma, the risk factors were as follows: for tumour recurrence, tumour size was larger than 5 cm and invasion occurred in subcutaneous fat; for distant metastases tumour localisation, tumour size was larger than 2 cm, and invasion occurred beyond subcutaneous fat as well as in the lymphovascular system. Risk factors for overall mortality include age, immunosuppression, tumour size larger than 2 cm, and lymphovascular invasion. Regarding the clinical outcomes, the same study reported that 96 of 319 patients died from the disease in a follow-up period with a median of 4.1 years [24].

Adequate resection is the mainstay of curative therapy in all cases discussed. Initial wide resection (2–3 cm) with clear margins leads to an excellent prognosis with low recurrence rates [25]. Limited or inadequate initial resection of DUPS due to a lack of reconstructive competence is not acceptable as this leads to high recurrence rates and eventually infiltration of the bone, as seen in advanced squamous cell carcinoma. Significant skin defects requiring local or microsurgical flaps might result from radical oncological resection. Adjuvant radio- or chemotherapy may be an adjuvant treatment option after radical surgery in these cases.

A clinical example is depicted in Figure 2.

### 2.4. Cutaneous Leiomyosarcoma

Cutaneous Leiomyosarcoma (LMS) is a rare, soft tissue neoplasm, often presenting as a solitary, firm, tender dermal nodule arising from smooth muscle cells in the dermis, m. arrector pili, or fatty tissue of the skin of the extremities, or the trunk, or the head and neck. Peak incidence is between 40 and 60 years of age.

Histopathologically, poorly delineated spindle-shaped atypical fascicles of myomatous cells merging into collagenous stroma are observed. Subcutaneous leiomyosarcomas are usually more sharply circumscribed [26].

Immunohistochemistry differentiates LMS from other spindle cell neoplasms as they express vimentin, desmin and smooth muscle actin [27]. It is important to distinguish between strictly dermal LMS and cases with subcutaneous invasion as the latter may be associated with local recurrence and/or systemic spread [28].

Distant metastasis is rare, but recurrence rates are high. Clinical long-term follow-up is therefore advised after complete surgical excision with wide margins.

### 2.5. Vascular Sarcomas

Vascular sarcomas described are Kaposi’s sarcoma (KS), primary angiosarcoma (pAS) and radiation-induced angiosarcoma (RIAS).

KS is associated with Kaposi sarcoma herpesvirus or human herpesvirus-8 (KSHV/HHV8). Four epidemiologic-clinical variants of KS are recognised: classic KS; endemic KS, the most aggressive AIDS-associated KS in HIV-positive patients; and finally, iatrogenic KS in immunosuppressed patients [29]. HHV8 is present in all forms. Multiple molecular mechanisms and host immunodeficiency precede KS development [30]. KS are purple or dark brown macular lesions, plaques or nodules, typically on the lower extremities. Classic KS usually has an indolent course, whereas endemic KS may take a fulminant aggressive clinical course with lymphedema. Histopathological findings are spindle cells and abnormal, dissecting vessels lined by thin endothelial cells [31]. HHV8 can be detected in cells by LNA-1 [32]. Local RT or chemo- and immunotherapy may be considered in more advanced cases. Systemic Doxorubicin is warranted in extensive cutaneous disease, visceral disease or rapid progression [33] and in AIDS-related KS [34].

Regarding angiosarcomas, pAS and RIAS are different clinical entities. pAS are very rare and aggressive neoplasms of endothelial origin, accounting for less than 2% of cutaneous sarcomas with an incidence of about 0.5 per 1,000,000 person-years in the United States [2]. They are typically located on the scalp or in the head and neck area of male patients who are older [35], presenting as rapidly growing and poorly demarcated bruise-like lesions. Haemorrhage or ulceration may also occur.

The pathogenesis of pAS is ascribed to the dysregulation of genes involved in angiogenesis and endothelial differentiation, including vascular-specific receptor tyrosine kinases [36]. Histologically, angiosarcomas are heterogeneous entities. Well-differentiated forms are characterised by irregular and anastomosing vascular structures lined by atypical endothelial cells in the periphery of the haemorrhage. The irregular vessels dissect through dermal collagen, distinguishing them from benign hemangiomas. Poorly differentiated AS are composed of solid sheets of spindle or epithelioid cells with mitosis, and lack definitive vasoformation, which can make diagnosis challenging. Immunohistochemistry for vascular markers CD31 and CD34 is useful [37,38].

Complete wide resection ± postoperative radiation is the gold standard for pAS. Surgery can be challenging due to ill-defined tumours with clinically undetectable intradermal spread [33,39]. In unresectable and metastatic disease, systemic Doxorubicin and taxanes are traditionally used [38,40]. Despite treatment, local recurrences are common and hematogenous dissemination occurs early. pAS carries a poor prognosis with 5-year survival rates ranging from 26 to –51% [35,41]. Lymphedema-associated AS, also known as Stewart–Treves syndrome, can arise from chronic lymphedema [42,43]. This is most often seen in patients with a history of breast cancer treated by radical mastectomy.

As mentioned above, RIAS are a specific subtype of vascular sarcomas. It is a rare complication to RT and occurs most commonly on the chest wall after breast cancer treatment (Figure 3). For primary RIAS cases (not recurrences), the current recommendation is not only wide excision with free margins but also excision of the entire radiation field [44]. This has significant benefits but requires extensive skin grafting, or if bone or other crucial structures are exposed, advanced and large-scale flap reconstructions. Unlike most sporadic angiosarcomas, radiation-induced AS and lymphedema-associated AS demonstrate MYC oncogene amplification, which is helpful in distinguishing radiation induced AS from other post-radiation conditions [45]. The low incidence and poor outcome for most of these patients of all the above-mentioned types of AS complicate suggestions of optimal treatment guidelines.

## 3. Treatment Algorithms and Strategies

### 3.1. Clear Margins Are Gold Standard

Pleomorphic dermal sarcomas comprise a group of rare, mesenchymal neoplasms arising from dermal or subcutaneous tissues. Tumours belonging to this entity display great heterogeneity regarding clinical and morphological presentations, challenging useful clinical guidelines. The initial presentation of the tumour may mimic more benign skin lesions, delaying prompt diagnosis. The most relevant and ultimate treatment, with regards to the general and local prognosis of PDS, is always complete surgical resection. Adjunctive therapy, e.g., RT, should not be used as a treatment modality in and of itself in cases with incomplete resection margins. Inadequate resection due to the fear of large defects or close relation to anatomically important structures causes significant morbidity for patients. This is not acceptable in light of the significant progress made in modern reconstructive surgery. Physicians in all fields should be aware of this as PDS has the potential to become aggressive, to cause local tissue destruction, to have high rates of recurrence, and to metastasise locally or distantly, which would subsequently require a highly specialised cooperation between oncological and surgical specialties for treatment. Interdisciplinary planning with a reconstructive microsurgeon is an integral part of the sarcoma board in reconstructive surgery cases. The challenge is to obtain acceptable aesthetic and functional outcomes based on an individualised assessment. Almost all knowledge on long term survival and surgical success after microsurgical reconstruction is based on soft tissue sarcomas as a general entity but generally reports excellent outcomes [46,47,48,49,50,51].

DFSP: Controversy exists regarding the surgical treatment of this type of dermal sarcoma. Some literature advocates for wide local excision to decrease locoregional recurrence [52] and others advocate for Mohs surgery [53,54,55]. In general, DSFP requires at least 2 and preferably 3 cm margins as the overall recurrence rate is reportedly about 50%; however, after adequate wide excision, this risk is reduced to 13% [56]. In light of high recurrence rates, temporary coverage of resection defects with split-thickness skin grafts followed by flap-based reconstruction for optimal functional and aesthetic outcomes after a recurrence-free interval of 24 months represents a promising strategy in selected patients. In one study, it was shown that cases with marginal excision showed a significantly higher recurrence rate than the wide excision group after long-term follow-up (median 65 months, range 31–190) [57]. In another study of long-term follow-ups (53 ± 36 months) after wide excision, the eight-year recurrence-free survival rate was 94% [58]. Facial tumours are an exception from this rule due to the clinically challenging localisation but should be discussed in the sarcoma board at the time of diagnosis. Close surveillance is suggested even beyond 5 years because late recurrences sometimes occur [52].

AFX/DUPS: As for all PDS, initial clear resection margins are the gold standard of surgery, as incomplete initial resection causes local recurrence [59,60]. No guidelines on surgical management and follow-up exist. In a recent study reviewing 100 published cases suggesting a probabilistic model, the appropriate surgical margins were found to be at least 2 cm for AFX and at least 3 cm for DUPS to clear 95% of all tumours [61]. Electrochemotherapy is a very recent potential treatment method for recurrent DUPS [62].

Dermal leiomyosarcoma: Consensus on the proper surgical treatment of dermal leiomyosarcoma does not exist, but guidelines advise early wide local resection [26]. Generally, an excision margin of 1–3 cm is recommended, including the fascia, securing radicality [63]. A large fraction of the knowledge on primary dermal leiomyosarcoma and its surgical management is based on retrospective studies, case series, single reports, etc. In a retrospective study on 48 dermal and 23 subcutaneous cases of LMS, treatment with wide local excision (minimum 1 cm margins) showed statistically significant lower rates of recurrences and metastasis compared to excision with narrow surgical margins. Fourteen cases of LMS treated by Mohs micrographic surgery had no recurrences or metastases [64]. Radio- and chemotherapy have no clinical benefits in LMS.

Primary angiosarcoma: Clear surgical margins correlate with better outcomes [65] and positive margins to poorer prognosis [66], which is why positive margins should always be managed by re-resection. Surgery can be challenging due to the close relation to anatomically important structures in the head and neck region [67]; however, pAS is treated by radical surgery—followed by RT [38,68].

RIAS: RIAS of the breast is, as mentioned above, a very aggressive condition. Due to the multifocal nature of this disease, clear resection margins are paramount in order to secure a local, recurrence-free survival [69]. The surgical strategy for RIAS depends on the clinical situation (primary versus recurrent cases). In primary cases, radical surgery and resection of the entire radiation field are promising [44,70]. However, in case of recurrence, radical surgery should be indicated with caution since the prognosis is poor anyway and patients usually experience the second recurrence shortly after extensive surgery. In a two-centre retrospective study including 31 patients undergoing initial radical surgery, two-thirds of the patients developed a local recurrence. The survival among those with local recurrence is better if the patient can undergo surgical treatment [71], but individual assessments are advised.

An overview of PDS and important clinical endpoints can be seen below (Table 1).

### 3.2. Adjuvant Therapy for Advanced Dermal Sarcomas

In clinical guidelines published by the American Joint Committee on Cancer, it is stated that the standard treatment of soft tissue sarcoma stages II or III is radical surgery (and adjuvant RT). Since the 1960s, RT has had a central role in adjunctive therapy for several distinct forms of soft tissue sarcomas. RT can be applied pre- or postoperatively, with the latter displaying less wound healing complications in a study on soft tissue sarcomas of the lower extremities [72], but there is limited evidence of its effectiveness. RT does not induce a survival benefit in cutaneous pleomorphic sarcoma [73].

It is noted that in cases of larger or high-grade tumours, adjuvant chemotherapy should be considered even though a clinically significant role for soft tissue sarcomas remains disputable. Anthracycline and Ifosfamide is recommended in combination as the preferred regimen [74]. Patients that are refractory to this standard treatment may try Docetaxel alone [75] or in combination with Gemcitabine as a second-line treatment [76]. In a randomised trial including 43 patients with high-grade sarcoma of the lower extremity, no differences in disease-free survival rates and overall survival were found between patients allocated to limb-sparing surgery plus adjuvant RT and patients undergoing limb amputation. Both groups received postoperative chemotherapy [77]. In another study, patients with high-grade soft tissue sarcomas of the extremities receiving adjuvant chemotherapy had improved disease-free survival compared to randomised controls [78]. These results were confirmed by DeLaney et al. [79]. A long-term (7 year) follow-up confirmed the sustained benefits of aggressive chemotherapy and surgery [80].

### 3.3. Biological, Targeted Therapy and Immunotherapy

Imatinib mesylate is an important addition to the treatment armamentarium of DFSP. In a phase II multicentre study including 25 patients, 36% were clinical responders after a 2-month preoperative daily administration of 600 mg of imatinib mesylate before wide local excision [81]. In a recent retrospective study of the long-term effects of neoadjuvant Imatinib mesylate or Pazopanib in 27 patients with locally advanced DFSP, 85% were disease-free after 50 months [82]. Pazopanib can be used in case of treatment failure by Imatinib.

Immunotherapy and new insights on the host/tumour microenvironment have in the last few years shown exciting treatment potential for soft tissue sarcomas in general, showing potential as adjuvant therapy for advanced primary dermal sarcomas as well [83]. In a clinical phase II trial, 200 mg of Pembrolizumab given intravenously showed a 40% objective response rate, assessed by CT or MRI imaging in advanced soft tissue UPS [84]. The response rate to PD1 inhibition may be dependent on the immunologic subtype [85]. In a retrospective analysis of seven patients with locally advanced or metastatic angiosarcoma, checkpoint inhibition by Pembrolizumab + Axitinib; AGEN1884, a CTLA-4 inhibitor, or Pembrolizumab alone indicated efficacy and even complete response in one patient with cutaneous angiosarcoma [86].

## 4. Conclusions

While new medical treatment strategies show promising therapeutic potential, radical surgery followed by individualised reconstruction is still the gold standard of treatment of primary dermal sarcomas and should not be compromised in favour of lesser surgical defects, e.g., in the facial region. The surgical planning of complex PDS can be challenging. Multidisciplinary cooperation in the preoperative planning involving reconstructive surgeons is paramount for such cases of extended and aggressive presentation. Advances in reconstructive microsurgery techniques offer functional and aesthetic acceptable outcomes. In cases with even more advanced local disease or systemic spread, adjunctive RT, chemotherapy or immune checkpoint inhibition are now available as an adjunctive treatment.

## Figures and Tables

**Figure 1 cancers-14-01693-f001:**
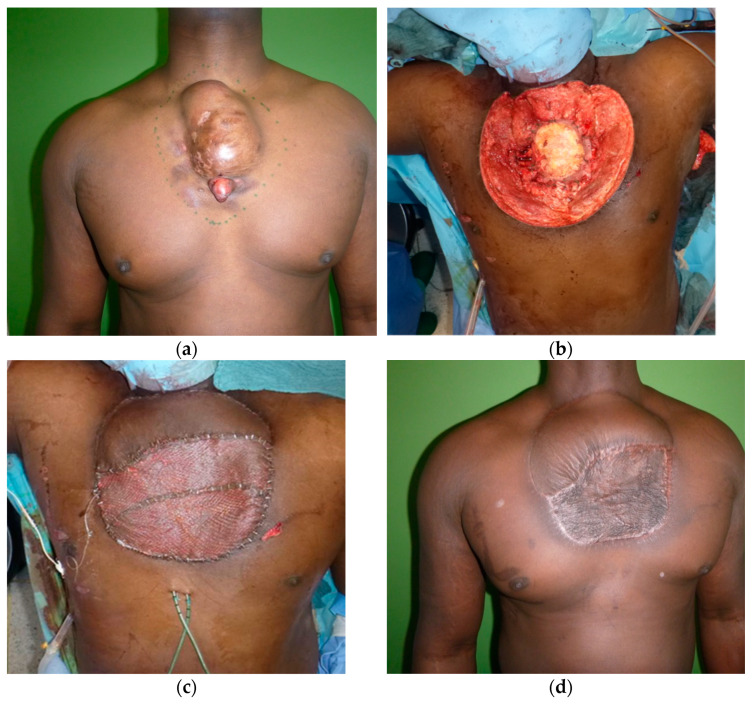
This figure illustrates a complex case of DFSP with infiltration to the sternum. (**a**) The initial appearance of the tumour prior to radiotherapy. (**b**) Defect after tumour resection with 3 cm margins and partial sternum resection. The defect measured 22 × 20 cm. (**c**) Defect reconstruction was achieved by use of a free conjoined parascapular and latissimus dorsi flap, which was anastomosed to the right internal mammary and vein in an end-to-end fashion [13]. (**d**) The result at 4-month follow-up.

**Figure 2 cancers-14-01693-f002:**
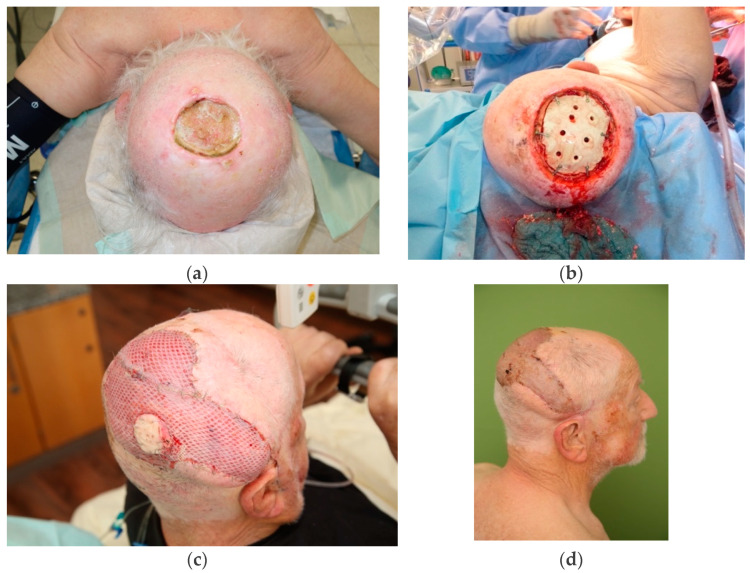
Microsurgical scalp reconstruction after resection of DUPS/AFX. Following an incomplete surgical resection and postoperative radiation of atypical fibroxanthoma on the scalp, this patient developed a local recurrence of DUPS/AFX 1 year after radiation therapy (**a**). The soft tissue and skull bone were resected with free margins in an interdisciplinary setting by neurosurgeons and reconstructive surgeons while the dura was preserved (**b**). The 9 × 10 cm bone-and soft tissue defect was covered by use of Palacos bone cement^®^ followed by a free latissimus dorsi flap (end-to-end anastomosis to the right temporal artery) and split skin from the thigh for cover (**c**). A photo of the result at follow-up at three months is displayed at the bottom right (**d**).

**Figure 3 cancers-14-01693-f003:**
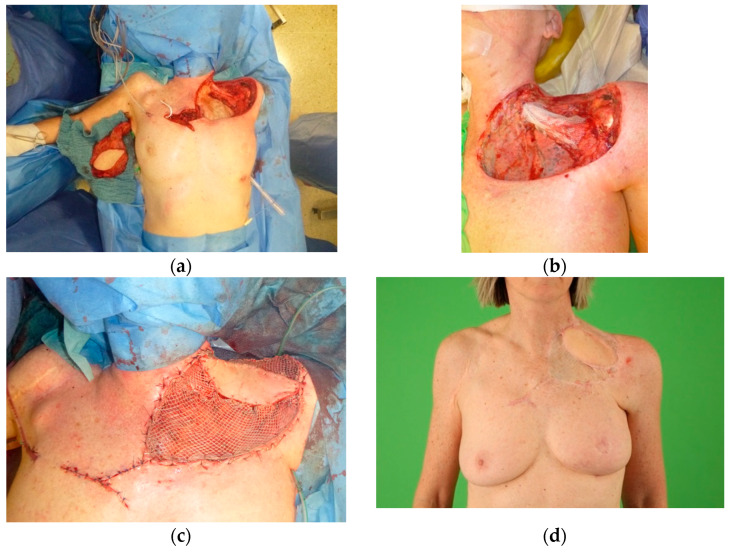
A case of cutaneous angiosarcoma (RIAS) of the left thoracic wall. The patient developed an angiosarcoma after radiation therapy targeting invasive, ductal breast cancer of the left breast. After complete resection of the radiation field including a part of the left clavicle and costa I (**a**,**b**), a free myocutaneous latissimus dorsi flap was raised and transposed, followed by an end-to-end anastomosis to the right internal mammary artery. Split skin was harvested from the thigh to cover the flap (**c**). Adjuvant chemotherapy (Doxorubicin, Adriamycin and Ifosfamide, and Dacarbazine) was given. After 6 months, the patient presented with a satisfactorily aesthetic and excellent reconstructive result (**d**).

**Table 1 cancers-14-01693-t001:** An overview of PDS and important clinical endpoints.

Type	Clinical Presentation	Pitfalls	Surgery	Adjuvant Therapy	Follow-Up
Dermatofibrosarcoma Protuberans	Painless pink or violet plaque	Fibrosarcomatous transformation Local destructive growth	Wide local excision: 2–3 cm incl. fascia ***	Imatinib	Every 6 months for 5 years *
Atypical Fibroxantoma	Ulcerative, rapidly growing lesion	Previous radiotherapy predisposes	Excision 2 cm	Radiotherapy	Every 6 months for 2 years
dermal undifferentiated pleomorphic sarcoma	Rapidly growing	Previous radiotherapy predisposes	Wide local excision: 3 cm incl. fascia ***	Radiotherapy Pembrolizumab	Every 3 months for 2 years + every 6 months for 3 years **
Leiomyosarcoma	Dermal nodule or diffuse growth	Attention to subcutaneous invasion High recurrence rate	Dermal: excision 1–3 cm Subcutaneous: excision 2 cm incl. fascia	Radiotherapy	Every 3 months for 2 years + every 6 months for 3 years
Kaposi sarcoma	Violaceous, purple/brown macular plaques/noduli	Can be mistaken for bruising or eczema	None, refer to dermatologic evaluation	Doxorubicin, Paclitaxel HIV-associated: antiretroviral therapy + chemotherapy	Individually planned
Primary angiosarcoma	Bruise-like lesions	Aggressive growth	Complete wide resection +/− postoperative radiation	Radiotherapy Doxorubicin	Individually planned, highly specialised
Radiation-induced angiosarcoma	Discolouration of the skin, papules	High recurrence rate	Excision of entire radiation field in primary cases	-	Individually planned, highly specialised

* MRI every 12 months; ** Grade 3 tumours: CT of the thorax + MRI after flap reconstruction every 6 months for 2 years and from there on every 12 months for 3 years; *** 3 cm in general and 2 cm at critical locations. Incl. = including.

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
