# Peer review of "A Multidisciplinary Approach to Complex Dermal Sarcomas Ensures an Optimal Clinical Outcome"

_cancers, 2022, doi:10.3390/cancers14071693_

Round 1

Reviewer 1 Report

Thank you for the opportunity to review this article.

The authors provide an overview of dermal sarcomas. The topic is interesting and I believe that a review would be useful to the literature as the heterogeneity of these lesions makes it difficult to find elements for a global assessment which is necessary for clinical practice.

The article is well written and the conclusions are consistent. However, I have a couple of suggestions for the authors.

1) The description of the various sarcomas almost completely lacks the histological aspects, despite histology may guide the differential diagnosis even before genetic/molecular analysis. In my opinion, it would be useful to focus on the histology of the individual lesions and the differential elements.

2) Little data is also provided on the clinical outcomes of the different tumours. What should we expect for these patients? I would like to see a more in-depth description of survival and recurrence rates, even according to specific treatments... or at least to highlight this data when it is already provided.

3) In table 1 it would be better in my opinion to have two different columns for "clinical presentation" and "pitfalls".

4) In my opinion it would be extremely didactic to provide a single pre-treatment clinical picture representative of the main dermal sarcomas.

Thank you.

Author Response

Reviewer#1: Thank you for your valuable input. Changes has been made to the manuscript accordingly.

1.The description of the various sarcomas almost completely lacks the histological aspects, despite histology may guide the differential diagnosis even before genetic/molecular analysis. In my opinion, it would be useful to focus on the histology of the individual lesions and the differential elements.

Answer:

Histopathology:

DFSP: the following sentence has been moved up (lines 81-82): ‘Histologically, DFSP is characterised by storiform, monomorphous spindle cell proliferation. CD34 positive cells can be used to determine DFSP from cellular dermatofibromas.’

AFX: This sentence has been added (lines 174-76): ‘The tissue is characterised by highly atypical cells with large nuclei, pleomorphism, hyperchromasia, and a high mitotic rate’

DUPS: This sentence has been added (lines 186-88): ‘Histopathology shows pleomorphic areas of anaplastic spindle cells, often mixed with bizarre giant cells in addition to an ordinary fascicular pattern of elongated smooth muscle-like cells’

LMS: This sentence has been added (lines 294-96): ‘Histopathologically, poorly delineated proliferation of spindle-shaped atypical myomatous cells arranged in fascicles merging into collagenous stroma. Subcutaneous leiomyosarcomas are more sharply circumscribed and typically include a vascular pattern.’

Regarding to angiosarcoma, please refer to the section, lines 323-25 and 337-42.

  1. Little data is also provided on the clinical outcomes of the different tumours. What should we expect for these patients? I would like to see a more in-depth description of survival and recurrence rates, even according to specific treatments... or at least to highlight this data when it is already provided.

Answer:

Clinical outcomes:

DFSP: please refer to line 122- 24: ’Overall, the 5-year survival of DFSP is 99.2%. However, insufficient initial surgery regarding radical excision with safety margins can cause poor prognosis and aggressive local tumour or bone invasion’

AFX: please refer to line 171: ‘that only rarely reoccurs or metastasises’.

DUPS: the following has been added to the text (lines 206-208): ‘Regarding to the clinical outcomes, the same study reported that 96 of 319 patients died from the disease in a follow-up period with a median of 4.1 years’.

LMS: Please refer to lines 311-312:Distant metastasis is rare, but recurrence rates are high. Clinical long-term follow up is therefore advised after complete surgical excision with wide margins’

Angiosarcomas: Please refer to lines 405-407: ‘The low incidence and poor outcome for most of these patients of all the above-mentioned types of AS complicate suggestions of optimal treatment guidelines’

  1. In table 1 it would be better in my opinion to have two different columns for "clinical presentation" and "pitfalls".

Answer:

This has been added to the table 1.

  1. In my opinion it would be extremely didactic to provide a single pre-treatment clinical picture representative of the main dermal sarcomas.

Answer:

We have provided clinical photographs before surgery as suggested for pt. 1 and 2. Patient 3 was referred to us after primary excision and unfortunately, photos are thus not available.

We also uploaded follow-up pictures of better quality to the manuscript (pt 1 and 3).

Reviewer 2 Report

The authors described the overwiev the dermal sarcomas adequatelly. I have only one remarks or motivatio. To authors

the TNM classification and staging of sarcomas depening the therapeutic approach. And or the application of the immune and bioogical therapy into table presented in the disscussion. 

the readers can recognise what therapy is adequate to the stadium and possibilty of radical surgery treatment and recomandation of therapy approaches.

Author Response

Reviewer #2:

Thank you for your valuable input. Changes has been made to the manuscript accordingly.

1.The authors described the overwiev the dermal sarcomas adequatelly. I have only one remarks or motivatio. To authors
the TNM classification and staging of sarcomas depening the therapeutic approach. And or the application of the immune and bioogical therapy into table presented in the disscussion. 
the readers can recognise what therapy is adequate to the stadium and possibilty of radical surgery treatment and recomandation of therapy approaches.

Answer:

Regarding to the question on TNM classification, we have added this sentence and reference to the introduction for clarification: ‘In Europe, the Fédération Nationale des Centres de Lutte Contre le Cancer (FNCLCC) grading system is generally used. It distinguishes three grades of malignancy including assessment of tumour differentiation, necrosis and mitotic count’ [1].

The option of immune therapy (Pembrolizumab) is added to table 1.

As primary dermal sarcomas as a group are very heterogenous, strict guidelines are challenging to obtain as discussed. An individual approach and thorough discussion in a highly specialised sarcoma board is advised, and this is added to the introduction. Please refer to lines 62-62 (‘However, as dermal sarcomas as a group are very heterogenous, strict guidelines are challenging to obtain. An individual approach and thorough discussion in a highly specialised sarcoma board is advised.’).

Round 2

Reviewer 1 Report

The authors have addressed all my comments. I therefore believe that the article is suitable for publication in its current form. Thank you.